# Shared community effects and the non-genetic maternal environment shape cortisol levels in wild chimpanzees

Patrick J. Tkaczynski [1,2,3,17✉], Fabrizio Mafessoni [1,4,17✉], Cédric Girard-Buttoz[1,2,5], Liran Samuni [1,2,6,7], Corinne Y. Ackermann[8], Pawel Fedurek[9], Cristina Gomes[10], Catherine Hobaiter[7], Therese Löhrich[11,12], Virgile Manin[1,2], Anna Preis[2], Prince D. Valé[1,2,13], Erin G. Wessling[6], Livia Wittiger[14], Zinta Zommers[15], Klaus Zuberbuehler[8], Linda Vigilant [1], Tobias Deschner[16], Roman M. Wittig[1,2,5,18] & Catherine Crockford[1,2,5,18]

Mechanisms of inheritance remain poorly defined for many fitness-mediating traits, especially in long-lived animals with protracted development. Using 6,123 urinary samples from 170 wild chimpanzees, we examined the contributions of genetics, non-genetic maternal effects, and shared community effects on variation in cortisol levels, an established predictor of survival in long-lived primates. Despite evidence for consistent individual variation in cortisol levels across years, between-group effects were more influential and made an overwhelming contribution to variation in this trait. Focusing on within-group variation, non-genetic maternal effects accounted for 8% of the individual differences in average cortisol levels, significantly more than that attributable to genetic factors, which was indistinguishable from zero. These maternal effects are consistent with a primary role of a shared environment in shaping physiology. For chimpanzees, and perhaps other species with long life histories, community and maternal effects appear more relevant than genetic inheritance in shaping key physiological traits.

[1] Max Planck Institute for Evolutionary Anthropology, Leipzig, Germany. [2] Taï Chimpanzee Project, Centre Suisse de Recherches Scientifiques, Abidjan, Côte d'Ivoire. [3] School of Biological and Environmental Sciences, Liverpool John Moores University, Liverpool, UK. [4] Weizmann Institute of Science, Department of Plant and Environmental Sciences, Rehovot, Israel. [5] The Ape Social Mind Lab, Institut des Sciences Cognitives, CNRS UMR 5229 Lyon, France. [6] Department of Human Evolutionary Biology, Harvard University, Cambridge, MA, USA. [7] Centre for Social Learning & Cognitive Evolution, School of Psychology & Neuroscience, University of St Andrews, St Andrews, UK. [8] Universite de Neuchatel, Institut de Biologie, Cognition Compare, Neuchatel, Switzerland. [9] Division of Psychology, University of Stirling, Stirling, UK. [10] Tropical Conservation Institute, Institute of Environment, College of Arts, Science and Education, Florida International University, Miami, FL, USA. [11] World Wide Fund for Nature, Dzanga Sangha Protected Areas, BP 1053 Bangui, Central African Republic. [12] Robert Koch Institute, Epidemiology of Highly Pathogenic Microorganisms, Berlin, Germany. [13] Unité de Formation et de Recherche Agroferesterie, Université Jean Lorougnon Guédé, Daloa, Côte d'Ivoire. [14] WWF Deutschland, Berlin, Germany. [15] Perry World House, University of Pennsylvania, Philadelphia, USA. [16] Institute of Cognitive Science, Comparative BioCognition, University of Osnabrück, Osnabrück, Germany. [17]These authors contributed equally: Patrick J. Tkaczynski, Fabrizio Mafessoni. [18]These authors jointly supervised this work: Roman M. Wittig, Catherine Crockford. ✉email: pjtresearchltd@gmail.com; fabrizio.mafessoni@gmail.com

In vertebrates, glucocorticoids (GCs) such as cortisol, secreted via the hypothalamic-pituitary-adrenal (HPA) axis, facilitate homeostasis via mediation of metabolic, immune, and behavioral responses to intrinsic and extrinsic stressors[1–3]. As a consequence of this multi-faceted and dynamic role, the regulation of HPA axis activation and GC secretion is of broad interest to ecologists and evolutionary biologists seeking to understand how animals adapt to changing environments[4–7]. Despite the flexibility of HPA axis activity in response to external and internal stimuli, numerous studies demonstrate consistent individual differences in HPA axis activity and reactivity to environmental stimuli[8]. Inter-individual variation in HPA axis regulation is linked to variation in immune function and can be predictive of variation in fitness outcomes[5,9–11]. For example, in both baboons and gray mouse lemurs, individuals with consistently elevated HPA axis activity have poorer survival outcomes and live substantially shorter lives than those with lower HPA axis activity[9,12]. Given the profound fitness effects of individual differences in HPA axis activity and regulation, understanding the relative role of genetics, experience, and environment in shaping these GC phenotypes is key to deciphering the evolution of physiological plasticity[5,13].

Results from human twin studies indicate that as much as 60% of the variation in cortisol levels may be explained by genetic effects[14–17]. While twin studies in humans have been important in revealing the genetic regulation of cortisol levels, they are often limited due to issues related to short-term sampling[18–20] and the restricted information available on the individual- and environmental-level factors that can become conflated within genetic effect estimates in human research[19,21,22]. Where the relative contributions of genes and environment have been assessed in relation to variation in cortisol levels or other health-related factors in humans, researchers often find a greater influence of shared family environments as compared to genetics (e.g. ref. [20]). However, these human studies are still often not able to delineate how much these familial effects are a product of parental effects or the macroenvironment in which families are situated, e.g. broader socioeconomic environments[18,20–22].

Non-human animal (hereafter animal) studies are less constrained than most human studies as researchers can either use experimental procedures, for example cross-fostering, or include appropriate environmental variables in their modeling to better account for shared environmental effects[23–27]. Like the aforementioned human studies, the animal literature suggests genetic inheritance plays a clear role in the formation of GC phenotypes, with, on average, 30% of the variation in cortisol patterns explained by genetic effects in these studies[23–27]. However, this work has largely focused on short-lived species[23,24,26,27]; (although see ref. [25]). Therefore, beyond human studies, we know comparatively little about how GC phenotypes emerge and are maintained in other long-lived species with protracted parental care. Determining the relative contributions of genetics, the microenvironment of parental effects, and the broader socio-ecological macroenvironment to variation in GC phenotypes is an important topic in evolutionary ecology. It can help us to understand the evolution of protracted development as a life history adaptation and the importance of non-genetic parental effects and plasticity within that extended ontogenetic phase.

Parental, and especially maternal, effects are recognized as major evolutionary drivers of phenotypic trait variation[28], including the regulation of cortisol and other GC levels[29–33]. In experimental rodent studies, maternal cortisol levels during pregnancy and during post-partum offspring rearing, as well as rates of maternal interaction with offspring, are all predictors of offspring cortisol levels and reactivity[29,31,34]. Rodent studies also suggest that maternal effects may occur via epigenetic processes,

such as DNA methylation of GC receptor promoter regions, leading to altered responsivity to stressors[29,32,34]. Primate studies of the role of maternal effects on cortisol secretion and reactivity have typically employed maternal deprivation paradigms, either via experimental separations or due to naturally occurring maternal loss[29,30,33]. Here, maternal loss is linked to elevations in cortisol levels or alterations to diurnal rhythm[30], however, these effects do not necessarily last into adulthood[30,33]. Similarly, in human studies, tests of maternal effects on cortisol regulation classically examine the consequences of negative maternal or early life circumstances (e.g. poor mental or physical health, low socioeconomic status, or maternal loss (reviewed in ref. [29])). Here, maternal loss or early life adversity related to maternal condition are associated with elevated HPA activity in offspring, which can last into adulthood for some individuals. Therefore, much of what we know about maternal effects on cortisol regulation in long-lived mammals is derived from studies of manipulated and/or extreme maternal circumstances.

In our study, we investigate the relative contributions of environmental, genetic, and non-genetic maternal effects (hereafter maternal effects) to variation in cortisol phenotypes in wild chimpanzees. Chimpanzees are a long-lived species, have a gestation period of approximately 8 months[35], and a prolonged immature dependency lasting at least 10 years[36], in which there is emerging evidence of maternal influences in growth, social development, survival, and future reproductive success[37–40]. Chimpanzees live in societies with a high degree of fission-fusion dynamics[41], and adults have relatively stable social phenotypes across years[42], with certain adult females, and thus mothers, consistently more gregarious and social than other individuals[43]. Maternal social characteristics are thus likely to be a key factor shaping the social environment of their offspring prior to adulthood. Therefore, during both pre- and post-natal phases, there is a long period in which maternal factors can shape endocrine phenotypes that may endure throughout adulthood in chimpanzees.

Using a dataset of 170 adult and immature individuals of both sexes from five different communities and two subspecies (western, *Pan troglodytes verus* and eastern, *Pan troglodytes schweinfurthii*), we applied well-established methods and mixed-effect models to partition variance and calculate the repeatability and heritability of cortisol levels in wild chimpanzees[44,45]. Repeatability coefficients allow us to quantify the level of individual variation in cortisol levels and thus whether it could be subject to selection[44], while heritability estimates indicate the relative contributions of environmental, parental and/or genetic factors in generating individual variation in this trait[45].

Trait repeatability is defined as the proportion of variance in a trait, here cortisol levels, explained by consistent individual differences[44]. In many wild animal populations, environmental heterogeneity can increase within-individual variation in GC levels and mask between-individual differences and thus trait repeatability[8,46–48]. As a consequence, more recent studies have begun to focus on the degree to which individuals vary in GC secretion in response to environmental gradients, i.e. GC reaction norms[13,49,50]. Under this framework, reaction norm repeatability refers to the proportion of variance explained by individual differences in average responses to the environment (reaction norm intercept) and in plasticity to the environment (reaction norm slope)[50]. Thus, heritability estimates of reaction norm intercepts and slopes reflect the relative contributions of environmental, parental and/or genetic factors in shaping average responses and plasticity to the environment respectively[51,52]. It is important to note that heritability estimates only reflect environmental/genetic/parental contributions to *among-individual* variation, i.e. a trait can have a genetic basis, for example, and a heritability estimate of 0 if the trait lacks variation in a population[53].

**Table 1 Summary statistics for final dataset used in the study.**

| | N individuals | N samples | Mean (±SD) N of samples per subject |
|---|---|---|---|
| All | 170 | 6123 | 36.02 (±48.17) |
| Adult males | 48 | 3243 | 67.56 (±79.97) |
| Adult females | 69 | 1742 | 23.86 (±19.72) |
| Immature males | 37 | 545 | 17.03 (±16.58) |
| Immature females | 32 | 593 | 15.95 (±11.09) |
| By pedigree | | | |
| Mothers with offspring in dataset | 19 | 648 | 34.11 (±24.20) |
| Fathers with offspring in dataset | 11 | 977 | 88.82 (±77.03) |
| Offspring with only their mother sampled | 31 | 1335 | 43.06 (±51.26) |
| Offspring with only their father sampled | 9 | 425 | 47.22 (±76.00) |
| Offspring with both parents sampled | 33 | 1096 | 33.21 (41.42) |
| Individuals with only maternal half siblings in dataset | 18 | 924 | 51.33 (±58.59) |
| Individuals with only paternal half siblings in dataset | 28 | 699 | 24.96 (±22.04) |
| Individuals with full siblings in dataset | 2 | 135 | 67.50 (±7.78) |
| Individuals with both maternal & paternal half siblings in dataset | 31 | 1467 | 47.32 (±58.23) |
| Individuals without relations in dataset | 62 | 1928 | 31.10 (±48.20) |
| By Population-group | | | |
| Taï-East | 33 | 1531 | 46.39 (±72.61) |
| Taï-North | 24 | 842 | 35.08 (±28.92) |
| Taï-South | 51 | 2470 | 48.43 (±56.58) |
| Budongo-Sonso | 45 | 1171 | 24.40 (±17.82) |
| Budongo-Waibira | 17 | 109 | 7.79 (±3.34) |

In total, 6123 urinary cortisol values from 170 individuals were included in the study. Note that certain individuals fall into several pedigree categories (e.g. an individual can be a father and have a maternal or paternal sibling), therefore, the number of individuals in pedigree categorization exceeds 170. The range of numbers of years of sampling of individuals in the dataset was 1–13 years, with a mean ± SD of 2.63 ± 3.01 years.

In chimpanzees, as well as humans, the circadian cortisol response is a well described reaction norm: levels rise gradually during sleep prior to a peak upon awakening, followed by declines throughout the day[54,55]. Across numerous chimpanzee cortisol studies, time of day (i.e. circadian rhythm) is one of the most consistent environmental predictors of cortisol levels[30,49,54,56–58], making it a promising reaction norm to investigate in terms of repeatability and heritability. In our study, we take a two-step approach, first establishing repeatability in circadian reaction norms, then examining heritability contributions to individual variation in these reaction norms. Chimpanzees are an interesting study species for this topic; although sampling across the lifespan is challenging for this long-lived mammal, many of the socio-ecological factors influencing variation in cortisol levels in this species are established[56,57,59–62]. Therefore, we can account and control for these factors in our statistical models, providing robust estimates of the relative contributions of genetic, maternal, and environmental effects to cortisol levels and circadian reaction norms in a wild, long-lived mammal with prolonged development and maternal care.

Given the prolonged maternal dependency observed in chimpanzees[37–40], coupled with evidence of genetic regulation of circadian cortisol responses in humans[14–17], we anticipated both genetic and non-genetic maternal effects to make substantial contributions to variation in cortisol patterns in wild chimpanzees.

## Results

Urine and fecal samples were collected from individuals of all life stages (2–53 years old) from five chimpanzee communities. For each urine sample ($n = 6123$ samples), we quantified cortisol levels using liquid chromatography-tandem mass spectrometry (LCMS[63];) and corrected for variation in water content in the urine using the specific gravity (SG) of each sample[64]. Therefore, we report urinary cortisol levels as ng cortisol/ml SG. From the fecal samples, we genotyped DNA extracts using a two-step amplification method including 19 microsatellite loci (per[65]).

In combination with behavioral observations of mother-offspring dyads, these genotypes allowed us to generate a pedigree containing 159 named mothers and 50 named fathers; 310 offspring had known mothers and 185 offspring had both known mothers and fathers. Following stringent criteria to measure circadian cortisol responses (see below), we included 170 individuals from this pedigree in our final dataset. Table 1 describes sampling by pedigree and group. In total, there were 64 mother-offspring and 42 father-offspring pairs sampled. Figures S1 and S2 illustrate the pedigree for individuals with urinary cortisol values in our study.

**Repeatability**. We used linear mixed-effect models (LMMs) with a Gaussian error structure to test adjusted repeatability in urinary cortisol concentrations, i.e. the proportion of variance attributable to between-individual differences given conditional effects[44,66]. Our key predictor of urinary cortisol concentrations (log transformed to achieve a symmetrical distribution) was time of day, which we converted into a continuous, hours-since-midnight value for each sample.

For our repeatability analysis, we fitted three models with identical responses and fixed effects but varying random effect structures (see Table 2 and "Methods" for full justifications of variables included). The *null* model included only the following random effects: a factor variable to account for variation in socioecology within groups ("group"), a factor to account for temporal variation within groups composed of group identity and the sampling year (termed "group-year"), and a variable to account for samples being pooled from various research projects ("project identity"). For the latter, although all projects follow the same protocol for sample collection and analysis, different projects may have different priorities for the individuals or contexts sampled, therefore, this random effect was included to account for such biases.

The *individual intercepts* model added random intercepts for individual identity and a dummy variable composed of individual identity and the sampling year (termed "ID-year"; used to

**Table 2 Structures of main models examining the repeatability and heritability of urinary cortisol levels and reaction norms in wild chimpanzees.**

| Model | Response | Fixed effects | Random effects & slopes | Model purpose |
|---|---|---|---|---|
| Null model | Urinary cortisol concentrations (ng/ml SG; log transformed) | - Demographic (five levels: adult male, cycling female, lactating female, juvenile male, juvenile female) × time of day<br>- Demographic (five levels: adult male, cycling female, lactating female, juvenile male, juvenile female) × time of day² | (1\| group)+<br>(1\| group-year) +<br>(1\| project identity) | Reference model to compare to *random intercept* and *reaction norm* models |
| Individual Intercepts model | | - Age at sample × time of day<br>- Age at sample × time of day²<br>- Seasonality × time of day<br>- Seasonality × time of day² | (1\| group)+<br>(1\| group-year) +<br>(1\| project identity) +<br>(1\| individual identity) +<br>(1\|\|ID-year) | To identify the contribution of consistent individual differences in average cortisol levels (random intercepts) to overall variation in cortisol levels |
| Individuals reaction norms model | | - Adult male-to-adult female ratio × time of day<br>- Adult male-to-adult female ratio × time of day²<br>- Community size × time of day<br>- Community size × time of day² | (1\| group)+<br>(1\| group-year) +<br>(1\| project identity) +<br>(time of day + time of day² \| individual identity) +<br>(time of day + time of day²\|\|ID-year) | To identify the contribution of consistent individual differences in plasticity to circadian effects (random slopes) to overall variation in cortisol levels |
| Heritability animal model | | - LCMS method (two levels: new, old) × time of day<br>- LCMS method (two levels: new, old) × time of day² | (time of day + time of day² \| group) +<br>(time of day + time of day² \| group-year) +<br>(time of day + time of day² \| project identity) +<br>(time of day + time of day² \| individual identity) +<br>(time of day + time of day²\|\|ID-year) +<br>(time of day + time of day² \| mother identity) +<br>(time of day + time of day² \| pedigree) | To determine the contribution of mother identity and pedigree (genetics) to overall variation in cortisol levels (heritability analysis) |

All models were tested using 6123 urinary cortisol values from 170 individuals.

compare within-year and between-year trait repeatability in urinary cortisol concentrations, see below). Lastly, the *individual reaction norms* model was identical to the *individual intercepts* model other than it included random slopes for the time of day within the random effects of individual identity and ID-year. Previous research found higher repeatability for the quadratic term of time of day in our study populations[49]; therefore, we included time of day as both linear and quadratic terms to model the potential circadian responses.

In all these repeatability models, we included as fixed effects variables previously shown to influence urinary cortisol levels (see Table 2 and "Methods" for full details). In order to best model the circadian response of cortisol levels, we included all fixed effects in interaction with the linear and quadratic terms of time of day.

We used a model comparison approach and leave-one-out cross validation[67] to examine whether the addition of random intercepts for individual identity (*individual intercepts model*) or random slopes for linear and quadratic effects of time of day (*individual reaction norms model*) improved the predictive power of each model. This would indicate whether individual differences in average cortisol levels or individual differences in cortisol plasticity to time of day, respectively, explained variation in cortisol levels well. We found strong support for the inclusion of the random intercepts for individual identity, but weak support for the inclusion of random slopes within-individual identity: in most model comparisons, the *intercepts only* models were preferred to the *reaction norm* models (Table S1), but with typically marginal differences in the expected predictive density. Given that our research interest lies both in the average cortisol levels and the circadian reaction norms, we included random slopes within-individual identity in our subsequent analyses.

Using established formulas for partitioning and comparing variances (see "Methods" for full details and equations[49,51,52]), from the *individual reaction norms* model, we calculated a within-year (i.e. short-term) trait repeatability estimate (variance explained by the ID-year variable) of 0.07 (95% credible intervals = 0.01, 0.13; Table 3) and a between years (i.e. long-term) trait repeatability estimate (variance explained by individual identity) of 0.04 (95% credible intervals = 0.01, 0.07; Table 3). We found substantial support for consistent individual differences in circadian reaction norm intercepts, i.e. average cortisol levels given the effect of time of day, with a reaction norm intercept repeatability estimate of 0.53 (95% credible intervals = 0.34, 0.69; Table 3). The mean reaction norm repeatability estimates for the linear and quadratic time of day slopes, 0.17 and 0.28 respectively, suggested a substantial proportion of variance in these phenotypes are attributed to individual differences. However, these estimates were associated with a large amount of uncertainty, with the lower credible intervals of both slopes close to 0.

The apparent lack of between-individual differences in circadian slopes was unexpected given the strong evidence for consistent individual differences in this phenotype in a previous study of adult male chimpanzees, a dataset which included individuals in our present study[49]. Therefore, to examine if the inclusion of adult females and immatures in our dataset contributed to uncertainty to our reaction norm repeatability slope estimates, we ran repeatability analyses for each separate demographic (adult males, adult females, immatures). The model structures were identical for each demographic with the exceptions that a continuous variable for dominance rank was included in the male model, a categorical variable (lactating or cycling) for reproductive status was included in the adult female model, and a categorical variable indicating sex was included in the immature model (we did not include dominance rank for the female and immature models due to insufficient dominance information for many individuals in these demographics; see

**Table 3 Repeatability coefficients (with 95% credible intervals) from reaction norm models quantifying circadian cortisol responses in wild chimpanzees.**

| Demographic | Coefficient | Estimate | (lCI, uCI) |
|---|---|---|---|
| All individuals combined | Within-year trait repeatability | 0.08 | 0.01, 0.13 |
| | Between years trait repeatability | 0.04 | 0.01, 0.07 |
| | Reaction norm repeatability for intercepts | 0.53 | 0.34, 0.69 |
| | Reaction norm repeatability for linear slope | 0.17 | 0.00, 0.75 |
| | Reaction norm repeatability for quadratic slope | 0.28 | 0.00, 0.94 |
| Adult males | Within-year trait repeatability | 0.06 | 0.01, 0.11 |
| | Between years trait repeatability | 0.02 | 0.01, 0.03 |
| | Reaction norm repeatability for intercepts | 0.40 | 0.10, 0.73 |
| | Reaction norm repeatability for linear slope | 0.26 | 0.00, 0.89 |
| | Reaction norm repeatability for quadratic slope | 0.27 | 0.00, 0.95 |
| Adult females | Within-year trait repeatability | 0.05 | 0.00, 0.11 |
| | Between years trait repeatability | 0.04 | 0.00, 0.10 |
| | Reaction norm repeatability for intercepts | 0.85 | 0.57, 0.99 |
| | Reaction norm repeatability for linear slope | 0.46 | 0.00, 0.99 |
| | Reaction norm repeatability for quadratic slope | 0.42 | 0.00, 0.98 |
| Immatures | Within-year trait repeatability | 0.10 | 0.01, 0.23 |
| | Between years trait repeatability | 0.04 | 0.00, 0.11 |
| | Reaction norm repeatability for intercepts | 0.43 | 0.00, 0.77 |
| | Reaction norm repeatability for linear slope | 0.25 | 0.00, 0.80 |
| | Reaction norm repeatability for quadratic slope | 0.18 | 0.00, 0.69 |

Repeatability coefficients were calculated across all individuals ($n = 170$), then within the specific demographics of adult males ($n = 46$), adult females ($n = 69$), and immatures ($n = 69$). Note that certain individuals ($n = 14$) appear both as adults and immatures in the overall dataset.

"Methods" for details). In the Supplementary Materials Figs. S3–S5 illustrate the circadian cortisol responses for each demographic and Tables S2–S9 provide the model summaries for the fixed and random effects of the *individual reaction norm* models of each demographic. For all demographics, we still observed a high amount of uncertainty for our reaction norm repeatability slope estimates (Table 3). Consistent with Sonnweber et al.[49], across and within demographics we generally found strong support for consistent individual differences in reaction norm intercepts. The reaction norm repeatability intercept estimates for adult males and females were clearly non-zero (Table 3); for immatures, although the estimate was high (0.43), the CI range was very wide, suggesting uncertainty.

**Heritability**. We estimated the heritability of urinary cortisol levels and circadian cortisol responses by implementing the well-established "animal model" approach[45], which estimates additive genetic variance and parental effects in a trait by partitioning variance in mixed-effect models. Our animal model was similar in structure to the *individual reaction norm*s model (see Table 2), with the major exceptions being the inclusion of the pedigree as a random effect to estimate additive genetic variance[28,45]. In addition, to partition the relative contribution of maternal effects (the main caregiver), we also included the identity of the mother of the individual sampled as a random effect. For this model, we also included random slopes for the linear and quadratic terms of time of day within all random effects (including project identity and group-year).

To ensure that we could effectively model heritability with the data available to us, we fitted three main different animal models. First, we fitted a full animal model using all the urinary cortisol values available to us (*Full heritability model*; 6123 samples from to 170 individuals). Second, to ensure any environmental or genetic effects were not an artifact of pooling samples from different populations and subspecies of chimpanzees, we fitted a model only including individuals sampled in the Taï forest (*Taï heritability model*; 4843 samples belonging to 111

individuals). Lastly, in male and female chimpanzees respectively, there tends to be a positive and negative correlation between dominance rank and GC levels[68,69], which should ideally be included in models to accurately model individual variation in cortisol levels. However, due to limited behavioral data for certain individuals (see "Methods"), we could not assign ranks to all individuals in the dataset. Therefore, we ran a third heritability model including only individuals to which we could assign ranks (*Dominance heritability model*; $n = 5691$ samples from 141 individuals).

For each of these models, having accounted for the influence of the fixed effects (see Table 2), we computed the proportion of variance attributable to *genetics* (calculated as variance explained by the pedigree), *maternal effects* (variance explained by maternal identity), and *shared community effects* (variance explained by the group variable). In addition to these factors related to heritability, we also calculated the proportion of variance attributable to temporal within-group factors (variance explained by the group-year variable), individual identity over the long-term (variance explained by individual identity), and the short-term (variance explained by the ID-year variable). For all these factors, we calculated the proportion of variance they explained for each component of the circadian reaction norm, i.e. average cortisol levels (intercepts) and cortisol responses to the linear and quadratic terms for time of day (*sensu*[51,52]). We also estimated the proportion of covariance between intercept, linear slope, and quadratic slopes explained by all these factors. Finally, we implemented a variation of the *Full heritability model*, referred to as *Trait heritability model*, in which the linear and quadratic circadian slopes were not present so that heritability could be calculated including the residual variance[51,52].

The relative contribution of our random effects to variation in circadian cortisol responses in wild chimpanzees from the *Full heritability model* are shown in Fig. 1, with a summary of the maternal and genetic effects in Table 4 (full details of all variance components are in Table S10 of the Supplementary Materials). Most of the variation in our models was explained by long-term inter-group differences (96.9%, credible intervals = 87.1, 99.2%),

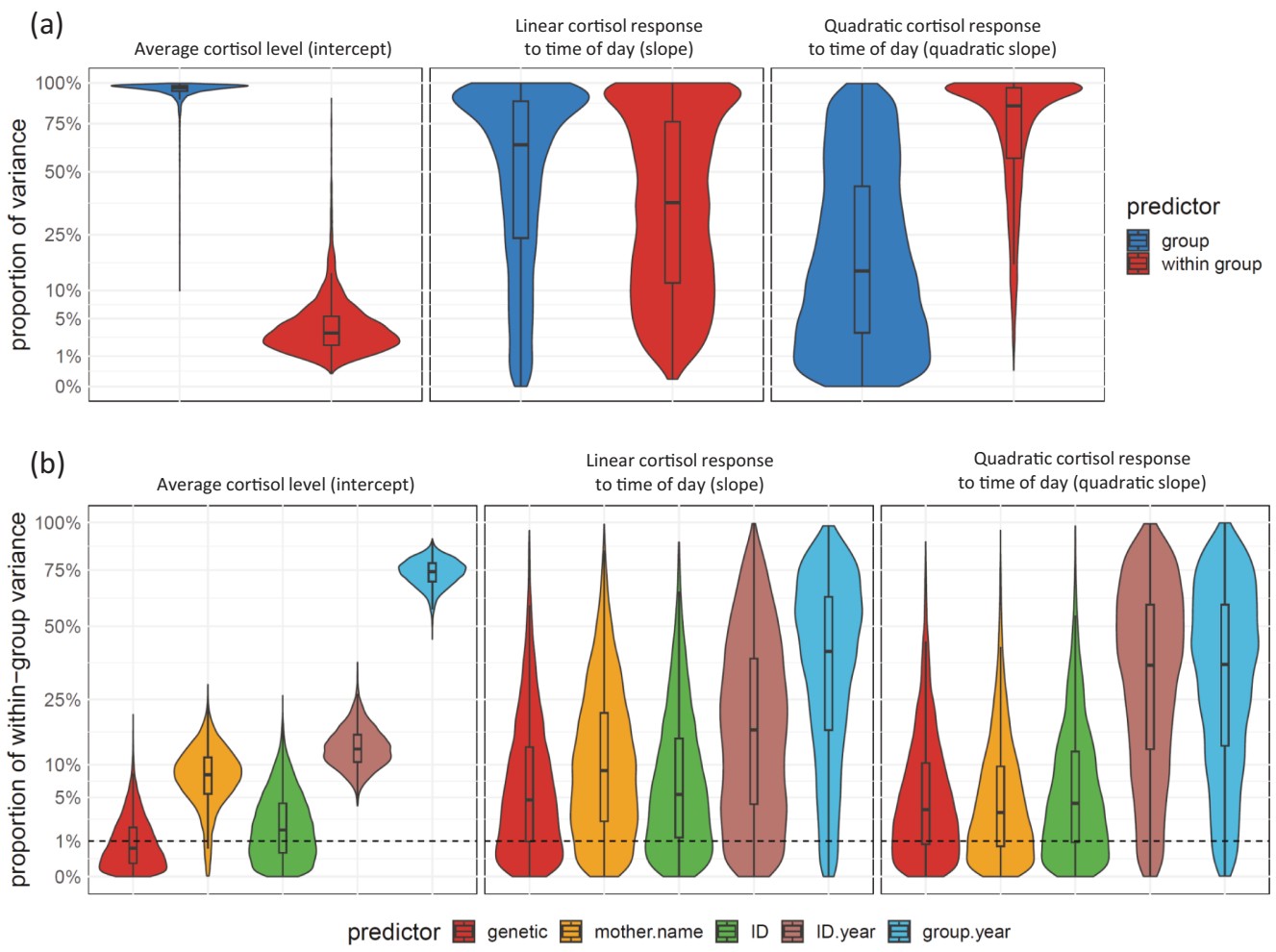

**Fig. 1 Estimates for the proportion of variance among the random effects in our model examining variation in circadian cortisol responses in wild chimpanzees. a** Proportion of variation explained by between-group versus within-group factors. **b** Proportion of the within-group variation explained by different factors. For both plots, the posterior distribution of the proportion of explained variance is shown as violin plots, with interquartile ranges represented by boxplots. The horizontal dashed line marks a proportion of within-group variance of 1%.

**Table 4 Summary of genetic, maternal, shared community group-year, individual-year and individual effect estimates (with 95% credible intervals) on circadian cortisol responses in wild chimpanzees.**

| Coefficient | Intercept | | Linear slope | | Quadratic slope | |
|---|---|---|---|---|---|---|
| | **Estimate** | **(lCI, uCI)** | **Estimate** | **(lCI, uCI)** | **Estimate** | **(lCI, uCI)** |
| Proportion of variance in components of circadian urinary cortisol reaction norms | | | | | | |
| Between-group effects (Shared community effects) | **0.97** | **(0.87, 0.99)** | *0.63* | *(0.01, 0.98)* | 0.15 | (0.00, 0.85) |
| Within group effects | **0.03** | **(0.01, 0.13)** | *0.37* | *(0.02, 0.99)* | 0.85 | *(0.15, 1.00)* |
| Proportion of within-group variance | | | | | | |
| Genetic effects | 0.01 | (0.00, 0.05) | 0.05 | (0.00, 0.38) | 0.04 | (0.00, 0.31) |
| Maternal effects | **0.08** | **(0.01, 0.16)** | 0.09 | (0.00, 0.49) | 0.03 | (0.00, 0.29) |
| Group-year effects | **0.74** | **(0.62,0.83)** | *0.40* | *(0.01, 0.84)* | *0.36* | *(0.01, 0.84)* |
| Individual identity effect | 0.02 | (0.00,0.10) | 0.05 | (0.00,0.41) | 0.04 | (0.00,0.35) |
| ID-year effect | **0.13** | **(0.08, 0.21)** | *0.17* | *(0.00, 0.70)* | *0.36* | *(0.01, 0.84)* |

Each coefficient represents a different component of the circadian cortisol response.
Coefficients in bold had genetic/maternal/group_year/ID_year/individual identity effect contributions with 95% intervals distinguishable from 0%, i.e. either with a credible interval larger or equal to 1%, or significantly higher than 0 according to the dataset permutation (Fig. 2). Italicized coefficients are those with lower credible intervals larger or equal to 1% yet associated with a high degree of uncertainty (interval range exceeds 80%).

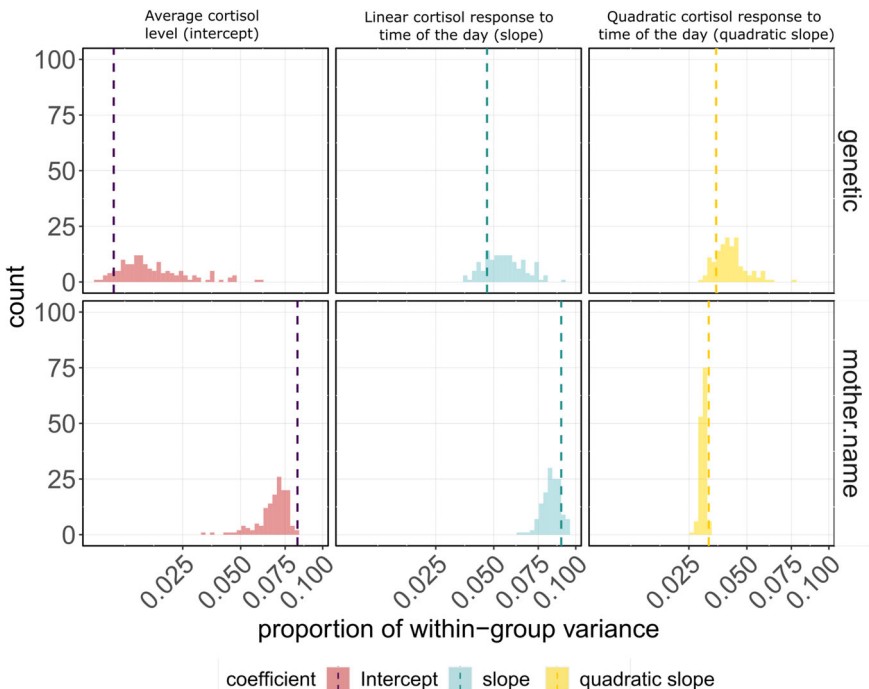

**Fig. 2 Comparison of genetic and maternal within-group variance estimates obtained from observed versus permuted data.** Median proportion of variance estimates obtained from the observed data are represented by dashed vertical lines; histograms represent the counts of each estimate value from 100 permutations. In this permutation analysis, the proportion of variance calculations includes all random effects, including our technical predictor, "project identity". Our final reported maternal effect estimate is higher than presented here as we consider only the biological predictors in that calculation. Figure S6 in the supplementary materials illustrates the permutations of all variance components in our heritability model.

particularly for the average cortisol levels, while the contribution of this predictor was less overwhelming and associated with high degree of uncertainty for circadian changes in cortisol (quadratic slope variance = 14.5%, credible intervals = 0.0, 85.4%). Maternal and genetic effects explained very little of the *overall* variation in cortisol, particularly for the average cortisol levels, however, when focusing on variation *within groups* only, maternal effects explained 8.3% (credible intervals = 1.2, 16.3%) of the average cortisol variation, while the genetic contribution was distinguishable from zero.

Similar estimates were obtained in the *Taï* and *Dominance heritability models* (Tables S11 and S12, Figs. S6–9). Both of these sub-analyses were informative about the results of our main model. Firstly, the strong contribution to variance of group in the Taï model confirmed that shared community effects were not the results of pooling results across two populations of chimpanzees. Meanwhile, the inclusion of dominance rank in the *Dominance heritability model* gave comparable results, with maternal effects higher than genetic effects, although the size of the maternal effect estimate was reduced to ~5% (Supplementary Table S12). This similarity in findings suggests that rank alone cannot fully explain the maternal effect observed in our main model.

Note, the credible intervals of our genetic and maternal effect estimates indicate a large degree of uncertainty (see Table 4). In addition, their values are bound to be positive as they were derived from the variance components of the random effects in the animal model. Hence, to assess whether maternal and genetic factors determine detectable non-zero effects and to test whether the differences between their variances could be due to chance, we performed re-sampling of the data and calculated the proportion of cases in which estimates were higher than for the observed data (i.e. false positives). Specifically, we reshuffled the identities of the individuals within their communities (and thus maintaining control of group-level environmental and social

factors) 100 times in the additive genetic matrix. Individuals newly classified as siblings after the permutation of the genetic matrix, were assigned to the same mother in the predictor "maternal identity", so that genetic relationships and maternal effects were always concordant. By doing this, we obtained permutations of the data that simulated genetic and maternal relationships expected by chance, while leaving unaltered the effects of all other predictors, keeping the same structure in the additive genetic matrix, and the same distribution of maternal relationships among individuals.

For the maternal effect on the intercepts (average cortisol levels), only one permutation out of 100 had estimates lower than the observed data (Fig. 2), while for the linear and quadratic slopes for time of day (cortisol circadian response), 14 and 0% of the permutations showed higher or equal values than the observed. This suggests that almost certainly the observed effects on the intercept and quadratic slope cannot be explained by chance and confirm a non-zero contribution of maternal effects to the cortisol phenotypes of wild chimpanzees.

We also used permutations to test whether the result that a higher proportion of within-group variance was explained by maternal rather than genetic effects could occur by chance. Only 1% of the permuted datasets and models generated a higher difference in variance estimates for maternal and genetic effects on average cortisol levels (i.e. intercepts) than that identified using our observed data and the *Full heritability model* (Fig. 3). Similar permutation results were also obtained for the *Taï* and *Dominance heritability models* (Figs. S10 and S11), as well as for a *Trait heritability model* (Table S13; Figs. S12–14), supporting the robustness of our results.

We conclude that the maternal environment is more influential than genetics in shaping cortisol responses in wild chimpanzees, while shared community effects are the clearest influence on this specific phenotype overall.

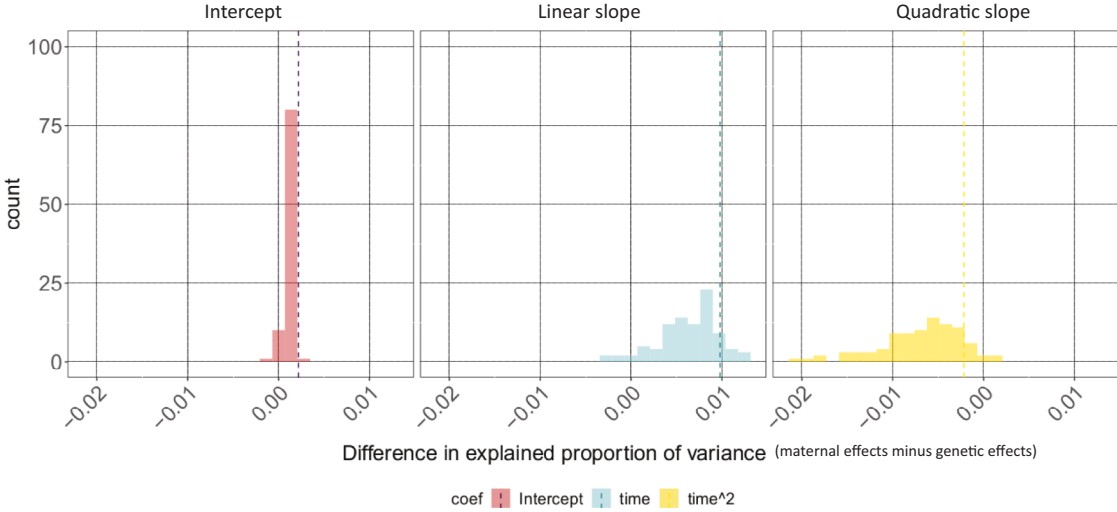

**Fig. 3 Differences in variance in cortisol phenotypes explained by maternal and genetic factors.** Estimates of the difference in the proportion of within-group variance explained by the maternal effect and that explained by genetic factors in the observed data (dashed line) and in 100 permutations of the data (histogram).

## Discussion

Our study leverages up to 18 years of long-term data collection of more than 6000 urine samples from 170 individuals to identify consistent individual differences in circadian cortisol responses in wild chimpanzees. Using this unique dataset, we find that group-wide, socioecological effects, represented by the "group" random effect in our models, have the most pronounced influence on variation in cortisol levels in this species. These effects held even when examining contiguous chimpanzee communities. Despite this predominant impact of socioecology, we were able to discern a significant role for the maternal environment in shaping average cortisol levels among individuals, with limited evidence for genetic effects alone driving individual differences. Our results were robust to different model structures and are corroborated by permutations of the data which indicate our maternal and genetic effect estimates are not artifacts of group structures. Further, the shared community effects cannot be explained by pooling samples from different populations and subspecies of chimpanzee, nor by established group-level predictors of cortisol such as group size or sex ratios. Our study shows the importance of long-term data collection in the wild, especially for long-lived species with protracted maternal care. It also raises important biological questions about the nature of both the macroenvironmental shared community effects and microenvironmental, non-genetic maternal effects that we documented, which combined accounted for virtually all variation, conditional on our fixed effect structure, in average cortisol levels in chimpanzees.

In our study, while much of the variation in urinary cortisol levels was attributable to long-term group effects, short-term group-level ("group-year" random effect) and individual-level ("ID-year" random effect) factors had a strong influence on cortisol phenotypes. These short-term effects illustrate the flexibility of this physiological phenotype in wild animals but also raise an interesting phenomenon of group-level, shared experience of temporal variation in cortisol levels in chimpanzees. In a similar recent study, heritability estimates for microbiome traits in a population of wild baboons (*Papio cynocephalus* and *anubis*) varied considerably within and between years based on seasonal dynamics and diet variation, highlighting the susceptibility of phenotypic variance, and thus heritability estimates, to shared environmental factors[70].

In chimpanzees, potential group-level stressors include food availability[62], dominance hierarchy instability[57,68], disturbance from neighboring communities of conspecifics[58], and disease outbreak[71]. The latter is unlikely to explain our finding as we excluded samples from individuals showing signs of illness (see "Methods"). However, we cannot exclude the possibility that the long-term community differences we observe in our study are in part driven by hangovers from extreme events, such as periods of high mortality from disease. Social bonds are key for cooperation in chimpanzees[72] and losing keystone individuals within cooperation networks could lead to long-term reduced within-group cohesion[73], in turn affecting cortisol levels across the entire group[74]. Instability could also arise if there are frequent switches in positions in dominance hierarchies and male rank instability has been shown to impact urinary cortisol concentrations in male Taï chimpanzees[57]. In most cases periods of rank instability tend to be short, lasting weeks or months rather than years, and would not necessarily explain the long-term group differences in cortisol levels in our study. Instead, social connectivity and social dynamics beyond rank may influence differences in group-level stress. We found a negative correlation between community size and cortisol levels (see Tables S2 in the Supplementary Materials). This association could be explained by the fact that although smaller communities may suffer more from inter-community competition[75], they could benefit from reduced within-community competition and improved cohesion and stability, as compared to larger communities in which individuals may struggle to maintain broad social networks[76]. One future avenue of research could focus in more detail on community differences in demographics and social cohesion as explanatory variables for the group-level differences driving cortisol variation in wild chimpanzees.

Regarding food availability and associated competition, the Taï communities are adjacent to one another, with relatively little between-group variation in the availability of food (see Fig. S15 for an illustration of between-group differences in urinary cortisol concentrations and food availability[77]). Therefore, we can tentatively discount this factor as the primary driver explaining between-group differences in cortisol levels. Chimpanzees are a highly territorial species[78], and the incorporation of more fine-scale behavioral data regarding competitive factors, e.g. rates of

incursion from neighboring communities[58,75], will also be beneficial in future studies on between-group variation in cortisol levels.

Decomposing the specific source(s) of group-level stressors in our dataset is an interesting and important avenue of future research. As average cortisol levels are predictive of survival[9,12], and between-group factors explain much of the variation in cortisol levels in our dataset, understanding the causes of group-level variation in cortisol levels may highlight risk factors for the long-term viability of certain chimpanzee communities or populations. More broadly, unlike most human studies to date, our focus on chimpanzees allowed us to fit highly robust models in terms of incorporating known socioecological predictors of cortisol levels in this species[57–59,61,62,68]. Furthermore, using a long-term dataset allowed us to examine cortisol levels both cross-sectionally and longitudinally, something often not possible in human research[18,21,79]. Therefore, our finding that community-level shared socioecology was the major predictor of cortisol levels in chimpanzees has some broader implications for our understanding about physiological plasticity in social, long-lived species, including humans.

Our results suggest a diminished role for heritable factors in shaping cortisol phenotypes for long-lived species. Remarkably, genetic factors had an even lesser role than the small but detectable effect of the maternal environment. In this species, males are the natal sex, while females typically disperse only once in their lifetime during adolescence[80]. Therefore, adult chimpanzees are socially bound to their communities, within which they exhibit complex within-group cooperation[72,78,81–83], in part to facilitate effective out-group directed aggression and in-group defense[78]. Although human cooperative tendencies can extend beyond an individual's immediate community, cooperation remains partially constrained by in-group favoritism, which can lead to substantial between-group competition[84]. In some recent human studies, community-level differences in stress exposure have been related to community-level variation in social inequality and culture[85,86]. These community-level discrepancies in the experience of social or physiological stressors were further illuminated by the covid-19 pandemic[87]. Our results suggest that community-level approaches to reducing excessive exposure to stressors could be more effective than approaches targeting single individuals, both in our closest living ancestors, and possibly in humans and other long-lived species.

Given the strong contribution of individual- and group-level temporal effects on variation in cortisol levels, it is notable that we were able to identify such a clear maternal effect on individual differences in our study, a key aim of our study. Although absence of evidence is not evidence of absence, the lack of a clear genetic effect in our results at least indicates a qualitatively stronger influence of maternal identity in shaping cortisol phenotypes in our study population. In a recent meta-analysis, Moore et al.[28] found a limited role for parental care in shaping the strength of parental effects on trait variation. However, as far as we are aware, few species included in the study demonstrate the prolonged mother-offspring association observed in chimpanzees. We hope that our results will encourage studies in other animals with protracted developmental phases or maternal associations to compare and contrast the relative influence of mothers and genetic inheritance. As with the macroenvironmental socioecological effects identified, determining the specific mechanism leading to the observed maternal effect merits further study, with factors such as protracted maternal care, epigenetic processes, or shared experience of stressors being possible explanations[29,88,89]. In wild chimpanzees, although maternal loss impacts later life reproductive success[37], there is no evidence that this is the result of long-term HPA axis activity alteration as effects on circadian cortisol patterns following

maternal loss do not endure into adulthood[30]. This time-limited nature of alteration of the HPA axis activity suggests that effects of adversity, and potentially maternal effects, are not explained by epigenetic mechanisms.

Chimpanzee offspring associate almost permanently with their mothers until around the age of 12 years[90], therefore, maternal social phenotype is the key determinant of the social environment of immature offspring. Previous research suggests maternal dominance status influences fecal GC levels in male, but not female immature chimpanzees[91], therefore, status alone is unlikely to explain the full extent of the maternal effect identified in our study (indeed we conducted a supplementary analysis for individuals to which we could assign dominance rank, suggesting some influence of dominance rank, but with a clear non-zero maternal effect independent of rank; see "Methods" and Table S12). Chimpanzee mothers can vary in their rates of direct social interactions with their offspring, i.e. demonstrate unique maternal styles[92]. Captive rodent studies show that rates of maternal affection can influence the regulation of the stress response in offspring, even inducing DNA methylation of GC receptor promoter regions[29,31,34]. Chimpanzee females also have distinct social phenotypes[42]. Therefore, the offspring of different mothers will have very different exposures to social behaviors and social partners. However, it is an open question whether these differences in exposure translates into variable learning of technical or social skills, such as extractive foraging[93], inheriting certain social relationships or components of their mother's social networks[83], as well as being primed to be more or less aggressive[42]. These social competitive factors may, in turn, influence HPA axis activity via nutritional or social stressor exposure[59,94,95].

In our study we find that the contributions of heritable, and particularly genetic, factors to cortisol phenotypes were low as compared to values reported in other analyses, such as human twin studies in which maternal and environmental effects cannot typically be measured (as highlighted in ref. [15]), or experimental animal studies employing less complex model structures (e.g. refs. [23,26]). The pedigrees used in these studies vary considerably: human twin studies often sample between 100–200 twin pairs and their parents[15], while certain animal studies can have over 1000 individuals within their pedigree (e.g.[24]). Despite drawing upon one of the largest datasets of its kind in a long-lived mammal (both in terms of number of individuals, samples, and years of study), our pedigree is comparatively small, although not uniquely so among animal studies (e.g. ref. [23]). Nevertheless, while the shallow nature of our chimpanzee pedigree can possibly explain the uncertainty of our estimates, it is unlikely to explain the reduced heritable contributions identified in our study. In our model, we included numerous socioecological predictors of cortisol levels in chimpanzees[57–59,61,62,68] as well as a random effect structure that could capture the relative contributions of shared community, genetic, and maternal effects. Our Bayesian approach, which allows us to capture uncertainty in our heritability estimates, and our permutation analyses, which suggests our results are not expected simply because of the structure of our pedigree, means that the qualitative and relative contributions of heritable components in our study remain illuminating. Thus, while in general, it is difficult to compare proportions of variance between different studies when model structures and complexities vary substantially[96], our study points to the fact that analyzing large datasets, encompassing different and diverse groups, could be key in revealing the relevance of different ecological conditions, shared community dynamics, and idiosyncratic group features in determining phenotypic traits. Moreover, based on results in other species, we do not contend that genetics have no influence on variation in cortisol in chimpanzees, only that any

genetic effect is challenging to discern due to the clearer role of maternal and group-level socioecological effects. Therefore, previous research, due to methodological limitations in effectively partitioning environmental and maternal effects, may have overestimated the importance of genetics in the formation of cortisol phenotypes in long-lived species.

In our study, we were able to show that average cortisol levels given circadian effects are repeatable across demographics, including adult females in various reproductive states and in immature individuals. However, unlike Sonnweber et al.[49], we only found strong support for consistent individual differences in average cortisol levels, rather than circadian slopes. This discrepancy is likely due to a combination of three factors. First, we included substantially more samples, individuals, sex, and age classes, despite using stricter criteria for individual inclusion. Second, out of necessity, different corrections for the dilution of the urine samples were used in the two studies (creatinine-corrected urinary cortisol in Sonnweber et al.[49]; specific gravity-corrected urinary cortisol in the current study; see "Methods"). Although urinary cortisol concentrations derived from each of these methods are usually correlated (and are in our dataset, albeit weakly), the ranges of values will be different, which may affect the degree of variation in quantified circadian slopes. Third, our models were inherently more complex in structure (e.g. the number of levels for each random effect and slope), which can make it difficult to compare repeatability between different studies[96]. We expect that our approach, necessary due to the question we are addressing and the size of the dataset, may not be optimal to evaluate the subtlety of individual differences in circadian slopes. Nevertheless, despite this discrepancy with Sonnweber et al.[49], across both studies, we find clear evidence of consistent individual differences in average cortisol levels, regardless of differences in sample sizes, model complexity, and correction factors used in calculating cortisol concentrations.

To conclude, in our study, the macroenvironment of shared community effects and the microenvironment of maternal effects are the major influences on cortisol variation throughout the lifespan in chimpanzees. The lack of a clear genetic influence on cortisol regulation in chimpanzees was surprising, given the weight of evidence in human twin studies, and suggests caution in interpreting human studies when non-genetic factors such as non-genetic maternal effects or community effects cannot be factored out[22]. Similar work on other ape populations, including humans, is vital to fully understand the relative contributions of genetics and non-genetic environmental effects on cortisol regulation in this taxon. Indeed, recent studies show that community-level differences in stress exposure are prevalent in human societies[85,86]. Our results show that experiences of differing group-specific stress levels may be a common aspect of our evolutionary histories. Overall, determining the precise mechanisms of both maternal and non-maternal community-level influences clearly merits further investigation and will contribute to our understanding of the role of society, parents, and developmental plasticity in shaping physiology in long-lived, social species.

## Methods

**Study site and subjects.** We used long-term behavioral, demographic, and physiological data collected between 2000 and 2018 from two field sites of two subspecies of chimpanzee. In Taï National Park (5° 52′ N, 7° 20′ E), Côte d'Ivoire, data were collected from three communities of western chimpanzees (East, North, and South[97];) and in Budongo Conservation Field Station, Uganda (2° 03′ N, 31° 46′ E), data were collected from two communities of eastern chimpanzees (Sonso and Waibira[98,99]).

In both Taï and Budongo, data on the chimpanzees are systematically collected by a combination of locally-employed field assistants and visiting researchers. Longitudinal data includes daily counts of group compositions, as well as recording of behavioral and social interactions using a combination of focal observations and ad-libitum sampling[100]. During observations of the chimpanzees, observers opportunistically collected urine and fecal samples from identifiable individuals. In Taï, regular observations of the chimpanzees commenced in 1990 (North, 1990–2018; South, 1999–2018; East, 2007–2018[97]) and regular urine sample collection (see below) commenced in 2000 (North and South, 2000–2018; East, 2003–2018). In Budongo, regular observations of the chimpanzees commenced in 1994 (Sonso, 1994–2018; Waibira, 2011–2018[98,99]) and regular urine sample collection commenced in 2005 (Sonso, 2005–2018; Waibira, 2017–2018).

**Urine sample collection and analysis.** We collected urine from identifiable individuals using a plastic pipette to transfer urine from the ground or vegetation into either a 2 ml or 5 ml cryovials. Cryovials were stored in liquid nitrogen once back in camp, typically within 12 h of collection. Frozen samples were transported packed in dry ice to the Max Planck Institute for Evolutionary Anthropology in Leipzig, Germany, where they were stored at ≤20 °C in freezers.

We quantified urinary cortisol levels for each sample using LCMS[63] and MassLynx software[101]. We used prednisolone (coded as "old method" in models, i.e. most samples analyzed prior to July 2016[63]), or testosterone d4 ("new method", i.e. all samples analyzed post September 2016[62]) as the internal standards. Samples analyzed using the "new method" tended to have higher urinary cortisol concentrations than those of the "old method", therefore, we included LCMS methodology as a fixed effect in our statistical analyses (see below). Intra- and inter-batch coefficients of variation for quality controls were 8.29% and 13.59%, respectively.

For each sample, we measured specific gravity (SG) using a refractometer (TEC, Ober-Ramstadt, Germany). SG values were used to correct cortisol measurements for variation in water content in the urine using the formula outlined by Miller et al.[64]:

$$SG_{corrected\ cortisol} = raw\ hormone\ concentration \times \frac{\left(SG_{population\ mean} - 1.0\right)}{\left(SG_{sample} - 1.0\right)}$$

The population means were derived from the samples included in this analysis. The SG population mean was 1.02 for Taï and 1.02 for Budongo.

**Fecal sample collection and pedigree generation.** Fecal samples were collected from identifiable individuals. The samples were collected using plastic bags and then either directly stored in ethanol, dried on silica gel, or using a two-step ethanol-silica method[102]. Dried samples were transported in silica to the Max Planck Institute for Evolutionary Anthropology in Leipzig, Germany.

Using these samples, microsatellite genotyping of DNA of 428 individuals ($n = 259$ for Taï; $n = 169$ for Budongo) has been conducted since 1999, with an average of 83% complete genotypes at 19 loci. Approximately 100 mg of each sample was extracted using either the QIAamp DNA stool (Qiagen) or the GeneMATRIX Stool DNA Purification (Roboklon) kits. We genotyped DNA extracts using a two-step amplification method including 19 microsatellite loci as detailed previously[65]. Using CERVUS 3.0 software[103], we compared the resultant genotypes using the 'identity analysis' function to confirm individual identities and the 'parentage analysis' function to confirm maternities and assign paternities, using confidence assessments of 80 and 95%. In the case of fathers, each of the paternity assignments received a high likelihood, and other potential sires (adult males present in the group at the same time) were excluded by two or more mismatches. In total, were able to reliably assign fathers to 43% of individuals represented in the dataset.

**Data preparation.** To provide an accurate measure of circadian patterns for each individual, we excluded certain samples where cortisol levels were expected to be elevated and not representative of normal circadian patterning. Here, we provide a detailed description of the sample exclusion process.

In female primates, including chimpanzees, cortisol levels vary with reproductive state[59]. Chimpanzee gestation is approximately 240 days[104]. Using demography data and the birth dates of offspring, we assigned females to three reproductive states (sensu[59]): pregnant (during the 240 days preceding the birth of any offspring), lactating (the 1095 days [based on average resumption of cycling in the population] subsequent to the birth of any offspring) and cycling (any other period of time when females were not assigned as pregnant or lactating). We included all adult female samples where we were able to assign reproductive state to the female at the time of sampling. Furthermore, following related studies[56,59], we excluded samples from pregnant females because cortisol levels tend to increase during pregnancy. In fact, interactions can occur between maternal and fetal HPA axes making it difficult to accurately determine maternal cortisol levels in isolation[105].

In immature chimpanzees (<12 years old), maternal separation elevates cortisol secretion and has short-term effects on cortisol circadian patterns[30]. Therefore, if immature individuals lost their mother prior to the age of 12 years old (social maturity), we excluded any sample collected from them following maternal loss during immaturity. However, as there is no evidence of long-term impacts of maternal loss in mature chimpanzees[30], all mature individuals were included

regardless of maternal loss during immaturity. Furthermore, injury and sickness can elevate cortisol levels and affect circadian cortisol patterns in chimpanzees[71]. At both Taï and Budongo, researchers systematically report any signs of illness of injury in the chimpanzees to onsite veterinarians. These reports are then validated and recorded by the veterinarians. Using these records, we excluded samples from individuals that displayed symptoms of sickness or injury on the day of sampling.

Lastly, in male and female chimpanzees respectively, there tends to be a positive and negative correlation between dominance rank and GC levels[60,68,69]. However, for one group in our study (Waibira), we had insufficient data to calculate ranks for the females, and in all groups, immature individuals are not traditionally considered part of the dominance hierarchy, and infrequently indicate submission to other immatures. Consequently, we were limited in the number of individuals we could confidently assign a dominance rank to, which would limit our sample size and pedigree depth for the heritability analyses. Therefore, in our main reported models, we did not include dominance rank as a fixed effect. Nevertheless, given the established importance of dominance rank in relation to stress in primates, we conducted a supplementary analysis. For mature adult males and females (excluding Waibira females, for whom we lacked sufficient data), dominance ranks were calculated using pant grunt vocalizations, a unidirectional call given from subordinate individuals[106]. We used a likelihood-based adaptation of the Elo rating approach to calculate ranks[107,108]; we assigned continuous Elo ranks to subjects for each day of sampling; each score was standardized between 0 (lowest rank) and 1 (highest rank) within each group. To be able to include immature individuals in our supplementary analysis, we converted ranks among adults into categorical variables: males and females with Elo rating equal to or greater than the mean on each day of observation were considered "high-ranking males" and "high-ranking females" respectively, those with ratings below the mean were considered "low-ranking males" and "low-ranking females" respectively. We then categorized immature individuals as either "offspring of high-ranking females" or "offspring of low-ranking females" based on the rank category of their mother on the day of observation. We took this conservative approach, as opposed to creating a linear hierarchy within immatures, as we have insufficient data within immatures to establish the hierarchy and a lack of evidence that offspring inherit rank in a linear fashion; indeed, anecdotal evidence suggests dominance among immatures is more likely to be driven by age than mothers. This new dataset included on 5,691 samples and 141 individuals; results from this separate heritability analysis using this dataset are reported in Table S12, once again finding substantially higher contribution of non-genetic maternal effects compared to genetic effects.

To ensure that we were able to characterize circadian cortisol patterns for each individual, we only included individuals with a minimum of 3 urine samples per year, collected during both morning and afternoon hours, such that the earliest and latest samples were separated by at least 6 h. The reason for this criterion was to be able to model a meaningful quadratic circadian slope for each individual within a given year of sampling[30].

To accurately model circadian patterns of cortisol for all individuals (our measure of cortisol reaction norm), we included interactions between the linear and quadratic time variables and all other fixed effects. We used 12 years of age to distinguish between adult (aged ≥12 years) and immature individuals (aged <12 years), as it is the age at which individuals socialize and forage predominantly independently from their mothers[90]. In addition to the demographic categorization (adult male, cycling female, lactating female, immature male, immature female) and age of each individual on the day of sampling, we included in the analysis a number of control variables known to influence cortisol levels. Both group size and mating competition[57,59,60] can affect GC levels in primates, therefore, we calculated both the number of adults (mean[+SD]; East 13.81[+2.19], North 8.92[+1.33], South 16.52[+2.58], Sonso 36.35[+4.12], Waibira 54.11[+2.50]) and the male-to-female sex ratio (mean[+SD]; East 0.35[+0.12], North 0.50[+0.19], South 0.37[+0.10], Sonso 0.49[+0.05], Waibira 1.02[+0.02]) at the time of sampling for each sample. Lastly, as seasonal variation in rainfall, temperature, humidity and food availability can influence cortisol levels in chimpanzees[62], we accounted for this circannual variation by converting the Julian date of sampling into a circular variable and including its sine and cosine in our models[61,62,109].

## Statistics and reproducibility

*Model fitting and verification.* All data preparation, models and analyses were performed using R version 3.6.3[110] and the RStudio interface[111]. Prior to testing our models, we applied the *vif* function of the 'car' R package[112] to linear model versions of our mixed models (i.e. lacking random effects) to test for any collinearity issues via examination of variance inflation factors (VIF). There were issues with collinearity if either "site" or "group" were included in the models as both variables were either collinear with each other or with "group size". Therefore, we retained just "group size", with all remaining VIFs <2.90. "Group" and "group-year" were also included as random effects to account for group-level confounds. To further test potential temporal batch effects, we ran additional models using a year-month random effect (one using all samples, another restricting our analyses to only data points for which data were available for more than one population). The results (Table S14 in the supplementary materials) were qualitatively and largely quantitatively identical to the less complex models presented in our main results.

All models were fitted with a Gaussian error distribution using the R package 'brms'[113]. For all models, numeric variables were standardized as z-scores. We fit models with weakly regularizing priors for the fixed effects ($\beta \sim$ Normal(0,1)) and for the random effects (student t-distributed (3, 0, 10)), with uniform (LKJ(1)) priors for covariance matrices of the random slopes. For all models, we specified four chains of 4000 iterations, half of which were devoted to the warm-up. Sampling diagnostics (Rhat <1.1) and trace plots confirmed chain convergence for all models. Effective sample sizes confirmed no issues with autocorrelation of sampling for all models. We further validated models with posterior predictive checks using the *pp_check* function of 'brms' (Figs. S16 and S17 in Supplementary Materials).

We estimated the heritability of urinary cortisol levels and their circadian patterning by fitting an "animal model", which estimates additive genetic variance in a trait by including the pedigree of individuals as a random effect[45]. Pedigrees were generated with the R package 'MasterBayes'[114]. The additive genetic matrix was computed using the *Amatrix* function of the R package 'AGHmatrix'[115].

*Repeatability and heritability calculations.* We calculated the repeatability and heritability of reaction norms following the same approach as references[45–48]. Specifically, we partitioned variance in average cortisol levels, $V_{intercept}$, variance in the linear cortisol response (slope) to time of day, $V_{linear}$, and the variance in the quadratic response (slope) to time of day, $V_{quadratic}$. These were computed as total variance excluding that explained by predictors related to technical aspects of the data (i.e. the project random effect).

For the repeatability analysis, total within-group variance in urinary cortisol concentrations ($V_{total}$) was calculated as the sum of variance explained by individual identity across years ($V_{individual}$) and within years, i.e. the ID-year variable ($V_{ID-year}$), as well as variance explained by group identity across years ($V_{group}$), and within years, i.e. the group-year variable ($V_{group-year}$).

Long-term trait repeatability (i.e. the proportion of variance in all urinary cortisol concentrations between years explained by individual differences) was calculated as:

$$V_{individual}/V_{total}$$

Short-term trait repeatability (i.e. the proportion of variance in all urinary cortisol concentrations within years explained by individual differences) was calculated as:

$$V_{ID-year} + V_{individual}/V_{total}$$

Reaction norm repeatability of average cortisol levels (i.e. the proportion of variation in average cortisol levels explained by individual differences) was calculated as:

$$V_{individual}/(V_{individual} + V_{ID-year})$$

Reaction norm repeatability of linear cortisol responses to time of day (i.e. the proportion of variation in the linear cortisol response to time of day explained by individual differences) was calculated using the variance of the random slope estimates for the linear term for time of day within the individual identity ($V_{linear,individual}$) and ID-year ($V_{linear,ID-year}$) random effects:

$$V_{linear,individual}/(V_{linear,individual} + V_{linear,ID-year})$$

Similarly, the reaction norm of quadratic cortisol responses was calculated as:

$$V_{quadratic,individual}/(V_{quadratic,individual} + V_{quadratic,ID-year})$$

For the heritability analysis and for each of $V_{intercept}$, $V_{linear}$, and $V_{quadratic}$, we partitioned the variance within groups ($V_{within}$) and between groups ($V_{group}$).

Hence, the proportion of variance explained by the shared community effects is:

Variance in average cortisol levels explained by shared community effects = $V_{group, intercept}/(V_{group, intercept} + V_{within, intercept})$

Variance in linear cortisol responses to time of day explained by shared community effects = $V_{group, linear}/(V_{group, linear} + V_{within, linear})$

Variance in quadratic cortisol responses to time of day explained by shared community effects = $V_{group, quadratic}/(V_{group, quadratic} + V_{within, quadratic})$

The within-group variance is defined, for each component of the reaction norm, as the sum of the variance explained by all biological predictors except group identity. Hence, for the *Full heritability model*:

Within-group variance in average cortisol levels ($V_{within,intercept}$) = $V_{genetic, intercept} + V_{mother, intercept} + V_{group-year, intercept} + V_{individual, intercept} + V_{ID-year, intercept}$

Within-group variance in linear cortisol responses to time of day ($V_{within,linear}$) = $V_{genetic, linear} + V_{mother, linear} + V_{group-year, linear} + V_{individual, linear} + V_{ID-year, linear}$

Within-group variance in linear cortisol responses to time of day ($V_{within,quadratic}$) = $V_{genetic, quadratic} + V_{mother, quadratic} + V_{group-year, quadratic} + V_{individual, quadratic} + V_{ID-year, quadratic}$

Thus, the proportion of within-group variance in the reaction norm components explained by genetic factors, henceforth reported as within-group genetic heritability, is then defined as:

Within-group genetic heritability of average cortisol levels = $V_{genetic, intercept}/V_{within, intercept}$

Within-group genetic heritability of linear cortisol responses to time of day = $V_{genetic, linear}/V_{within, linear}$

Within-group genetic heritability of quadratic cortisol responses to time of day = $V_{genetic, quadratic}/V_{within, quadratic}$

A similar formulation applies for the variance explained by the maternal identity, henceforth defined as maternal effects:

Within-group maternal effects on average cortisol levels = $V_{mother, intercept}/V_{within, intercept}$

Within-group maternal effects on linear cortisol responses to time of day = $V_{\text{mother, linear}}/V_{\text{within, linear}}$

Within-group maternal effects on quadratic cortisol responses to time of day = $V_{\text{mother, quadratic}}/V_{\text{within, quadratic}}$

We also implemented a *Trait heritability model*, in which variance is not stratified in reaction norm components, hence unexplained residual variance $V_{\text{residual}}$ can be considered in the total variance.

In this case, heritability is defined as:

$$V_{\text{genetic}}/(V_{\text{genetic}} + V_{\text{mother}} + V_{\text{group-year}} + V_{\text{individual}} + V_{\text{ID-year}} + V_{\text{residual}})$$

**Ethical approval**. All methods were non-invasive and were approved by the Ministries of Research and Environment of Côte d'Ivoire, and Office Ivoirien des Parcs et Réserves. All aspects of the study comply with the ethics policy of both the Max Planck Society and the Department of Primatology of the Max Planck Institute for Evolutionary Anthropology, Germany (www.eva.mpg.de/primat/ethical-guidelines.html) for the ethical treatment of non-human primates.

**Reporting summary**. Further information on research design is available in the Nature Portfolio Reporting Summary linked to this article.

## Data availability

All data used in the analyses are available via Figshare (https://doi.org/10.6084/m9.figshare.13720765.v1)[116]; all code used in the analysis are available at: https://github.com/fabrimafe/cortisol_heritability.

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

## Acknowledgements

We thank the Ministère de l'Enseignement Supérieur et de la Recherche Scientifique, the Ministère de Eaux et Fôrets in Côte d'Ivoire, the Office Ivoirien des Parcs et Réserves, the Uganda Wildlife Authority and the Uganda National Council for Science and Technology for permitting the study. In Côte d'Ivoire, we are grateful to the Centre Suisse de Recherches Scientifiques en Côte d'Ivoire and the staff members of the Taï Chimpanzee Project for their support. In Uganda, we thank the management and staff of the Budongo Conservation Field Station. We are indebted to the efforts of Christophe Boesch and Vernon Reynolds in the establishments of the study field sites and their contributions to years of data collection. We also thank the many field and research assistants that help generate the data for this project. We are extremely grateful for the work conducted in the laboratories of Tobias Deschner and Linda Vigilant in the Max Planck Institute of Evolutionary Anthropology, Leipzig, Germany, specifically the efforts of Róisín Murtagh, Vera Schmeling, Janette Gleiche, Anette Nicklisch, Juliane Damm, Carolyn Rowney, and Jared Cobain. We also thank Ruth Sonnweber and Verena Behringer for useful discussions on the topic. This study was funded by the Max Planck Society and the European Research Council (ERC) under the European Union's Horizon 2020 research and innovation program awarded to C.C. (grant agreement no. 679787). L.S. was supported by the Minerva Foundation, C.Y.A. and A.P. received funding from the LSB Leakey Foundation, C.Y.A. also received funding from Subvention Egalité (University of Neuchâtel, Switzerland) and Fonds des Donations (University of Neuchâtel, Switzerland). C.G. was supported by the Wenner-Gren Foundation. VM was supported by a grant of Deutsche Forschungsgemeinschaft (DFG) granted to R.M.W. (WI 2637/3-1). Core funding for the Taï Chimpanzee Project was provided by the Max Planck Society since 1997 and for Budongo Conservation Field Station by the Royal Zoological Society of Scotland since 2008.

## Author contributions

C.C., C.G.B., F.M. and P.J.T. conceived the study. A.P., C.C., C.G., C.G.B., C.Y.A., E.G.W., L.S., L.W., P.F., P.J.T., P.D.V., R.M.W., T.D., T.L., V.M., and Z.S. collected data. T.D., L.S., C.H., K.Z., C.C., and R.M.W. provided long-term data. P.J.T., F.M., C.C., C.G.B., P.F., T.D. and R.M.W. helped design the study; F.M., C.G.B. and P.J.T. performed the statistical analyses; T.D. oversaw the laboratory analyses; L.V. supervised and conducted genetic parentage analyses; P.J.T. wrote the first draft of the manuscript, all authors contributed to subsequent editing.

## Funding

## Competing interests

We have no competing interests to report.
