## [Peer Review File · Communications Biology]

Reviewers' comments:

Reviewer #1 (Remarks to the Author):

In this study, the authors addressed associations between genetic effects, non-genetic maternal effects, and environmental effects, and cortisol levels in wild chimpanzees. The authors found a significant contribution from between-group effects (shared community effects). When evaluating within-group variation, non-generic maternal effects accounted for some of the individual differences, while genetic effects were negligible. They conclude that (short- and long-term) ecological processes are more important in shaping cortisol levels than genetic ones. I think these findings are of high interest to the general readership and provides new insights into the cortisol dynamics in long-lived animals.

The manuscript is well written with vast details on their methods to allow both repeatability by others, as well as a fair judgement of the reader. Although I am familiar with the main topic of the study, I am not an expert on the wet lab analyses carried out, so my main comments are about the analysis interpretation and potential limitations.

First, the dataset is good with a large amount of urine samples. Yet, the number of parent-offspring pairs seems relatively small (below I also ask to clarify these numbers in the narrative). I think it would be good to discuss the methodological limitations of having low number of pairs in trying to answer questions about heritability. Are there other references validating similar models across sample sizes? In other words, can the study be underestimating the contribution of genetic effects? This is important as several other references cited by the authors suggest the opposite.

On the other hand, the authors mention they discarded offspring whose mother died in their analysis. I understand why they did it and I think it is a good approach. Now, it would be interesting to discuss/predict whether the contribution from between-group effects and even the contribution from group-year effects found has something to do with group-specific demographic rates (i.e., sex- and age-specific mortality). For example, if a portion of the population structure in each site presents higher mortality risks, do the other members (those in the analysis) respond to this in term of their cortisol levels? Is the social structure after an individual dies significantly disrupted and could this affect cortisol variation at the community level? The authors mention potential ecological stressors but none of them address demographic rates and potential social disruption directly. On this same topic, can other dimensions of sociality (social integration, connectivity) be a major contributor of the observed cortisol variation, rather than just rank? I think mentioning this is worth and can lead to future questions.

Below minor comments for each section of the manuscript.

Introduction:

I don't think the first paragraph needs to be separate from the second. To me, it would read better if they were together.

Lines 62-63. This is very interesting and important for your claim. I suggest you add one or two other references from other primate populations to make the statement stronger. I know there are several examples from the Cayo Santiago macaques. There are probably other references from other populations as well.

Lines 89-93. This sentence is a bit complicated. I suggest splitting it into two.

Materials & Methods.

Lines 687-690. Is this health data collected as part of the systematic data? If yes, clarify this in lines 618-620. If not, clarify here how this was collected.

Line 771. If ID is the identity of the individual, then is the subscript "individual" the same as "ID"? I find it a bit confusing to use two different symbols for the same variable. Clarify.

Results.

Lines 177-181. You mention 310 offspring with mothers and 185 with father and mother. Then you reduce the total sample size to 170. In Table 1, you mention that 31 offspring had their mother sampled, 9 their father, and 33 both parents. Can you clarify here (after mentioning the 310 offspring and 185 offspring) how many parent-offspring were finally included in the analysis?

Lines 243-244. Can you clarify here that you did not include random slopes for individual identity?

Discussion.

See my main comments above.

Reviewer #2 (Remarks to the Author):

In this paper, the author(s) use an impressive sample size ($n=170$) for a wild-long lived primate to demonstrate that circadian stress phenotype, measured from urinary cortisol assays, shows some consistency within individuals but differs between groups. Moreover, the authors provide evidence that maternal effects but not genetics are associated with differences in within group stress phenotype. They conclude that community and maternal effects drive plasticity in stress phenotype. The authors rigorously assess the robustness of their main findings using several strategies. First, based on longstanding work in this system, they have substantial prior knowledge to inform their models, and thus, control for potential confounding. Next, they use sensitivity analyses, including an analysis of single social group and in models that include dominance rank to assess the stability in the direction, magnitude, and precision of their estimates of interest. Finally, they use permutations of the animal IDs that generated random genetic and maternal relationships to assess possible type I error. I found the analysis to be thoughtful and rigorous. I also wanted to note that I think the authors' description of the distinction between traits having a genetic basis and being heritable (lines 134-148) is important, clear, and likely of great use to potential readers before they view the results. Finally, the paper is well written, to the point that many of the questions I formulated when reading through the results were already addressed in the discussion. This was a joy to read. In addition to these broad sentiments, below I pose a few questions and provide some (minor) suggestions that I hope improve the manuscript.

Questions:

1. Given the large proportion of variance explained by community, or socioecological macroenvironmental conditions, could the authors provide some information and discussion about 1) dynamics of social hierarchies in this species and 2) possible effects of discordant timing of sample collection. More specifically, it is noted in the introduction that individual social phenotypes are stable (lines 119-120), but are dominance hierarchies relatively stable in these groups when the data were collected? Controlling for community size and the sensitivity analysis, in which rank is included in the model is useful and may be the best solution given the data, but whether being high or low rank is stressful per se likely depends on the context in which that rank is achieved. Second, could between group differences reflect the timing of when data were collected since data collections were, to some

extent, non-overlapping? Even among the Tai groups the study periods are variable. Is it possible that there are cohort effects that drive the large between group differences in variance and is there any concern that this may cause an underestimation of the genetic effect?

2. I found the results in which maternal ID accounted for a non-zero variance in stress phenotype fascinating. Considering that maternal loss does not appear to have an enduring effect on adult stress phenotype and that after controlling for rank maternal identity still accounts for some of the variance in stress phenotype, do the authors have other ideas about what may underly this pattern...shared social networks between mothers and offspring or behavioral differences in maternal care? They briefly address this in the discussion on lines 557-562, though I thought it would be nice to expand on these ideas as space permits e.g., do offspring of more strongly connected mothers have lower or less reactive stress phenotypes in this species? Slightly more explanation here could help tie the results back to concepts about maternal effects and stress phenotype development in captive rodents and primates that appear in the introduction.

3. The lack of variance in stress phenotype explained by genetics is also interesting, especially considering prior work in human twin studies. In reconciling these discrepant findings, the authors note that human studies often lack the same detail in measuring environmental factors as studies of wild animals. Do they have any thoughts about how the assignment of paternity to 43% of the study sample may, if at all, relate to differences in variance explained by genetics?

Minor Comments

1. Line 86: Is describing parental care as 'short in duration' appropriate considering the species-specific life histories and given that some small animals have relatively quick development and short lifespans? In either case, I don't think post-hatching/post-parturition parental care is absent in any of the species studied in the references cited.

2. Lines 98-101: For a more nuanced presentation of the maternal effects on GR DNA methylation and downstream stress physiology, the authors may consider citations that failed to reproduce the work of Meaney et al., i.e., where low licking and grooming is associated with higher GR DNA methylation. For example, Fleming et al., (<https://doi.org/10.1037/bne0000014>) find high licking and grooming is associated with higher GR DNA methylation.

3. Some of the italicized text in table 4 does not match the table description. For example, the CI for ID-year estimate for quadratic slope is not italicized but the ICI is 0.01. Also, do the authors have a specific threshold for 'large range' CI used to distinguish bold vs italicized text?

4. For Figure 1, I note a few minor points that may help improve interpretability. 1) For the within-group panel b, having the explanatory variables listed in the same order as in table 4, would be useful. 2) There appears to be differences in the ICI error bars even when those are supposed to equal 0. Maybe this is a visual artefact of image quality or possibly a null reference line may help here. 3) Table 4 and Figure 1 appear redundant. Personally, I found table 4 to be more informative.

5. For figure 2, the dashed line for the median observed variance is hard to read, especially where it intersects the distribution of estimates from the permutations.

6. Lines 490-494, this is an interesting negative association between group size and cortisol. While not critical, I'd be interested to know if dominance hierarchies are more stable in larger groups?

Reviewer #3 (Remarks to the Author):

This paper aims to determine the contribution of genetics, parental effects, and the socioecological macroenvironment to variability in cortisol phenotypes in populations of chimpanzees over 18 years. The major claims of the paper are that the largest proportion of variability is explained by the macroenvironment and secondary to that, the microenvironment. This is novel and exciting data for the field. As the authors rightly pointed out, these studies have not been done in a long lived species with protracted parental care. These studies have not been carried out in humans either without some extreme conditions that invariably will have major influence on the HPA axis. Given this, the

contribution of genetics may have been overestimated in previous studies. As someone who has studied the HPA axis and stress reactivity throughout my career, this paper will affect how I think about the phenotypic variability associated with cortisol as we look at the relationships between parents and their children. The manuscript is logical and well written. I have only minor suggestions.

Discussion

Lines 585-588 - I am not convinced that the urine correction (specific gravity vs. creatinine) between the current study and Sonnweber et al., 2018 would lead to different results as the majority of studies have shown that these two measures are highly correlated.

I am fascinated that this paper shows that socioecological effects have a greater impact on cortisol phenotypic variation compared to parental factors. While some examples of these factors are presented in the discussion, I am wondering if the authors can speculate how these macro environmental effects could potentially translate to humans since the ecological conditions are not present in the same way as in the chimpanzees.

Response to reviewers

25th March 2023

Dear Reviewers,

We are extremely for your feedback and comments on our submission. Below, we respond in bold text to each reviewer comment, and provide line references for changes made in the manuscript in relation to these comments. We have submitted one version of the manuscript with changes highlighted in yellow, and one formal version of the revision without these highlights.

These edits and your feedback have made the paper stronger, bringing more clarification to the interpretation of the results.

Sincerely,

Drs Patrick Tkaczynski & Fabrizio Mafessoni

Reviewers' comments:

Reviewer #1 (Remarks to the Author):

In this study, the authors addressed associations between genetic effects, non-genetic maternal effects, and environmental effects, and cortisol levels in wild chimpanzees. The authors found a significant contribution from between-group effects (shared community effects). When evaluating within-group variation, non-generic maternal effects accounted for some of the individual differences, while genetic effects were negligible. They conclude that (short- and long-term) ecological processes are more important in shaping cortisol levels than genetic ones. I think these findings are of high interest to the general readership and provides new insights into the cortisol dynamics in long-lived animals.

The manuscript is well written with vast details on their methods to allow both repeatability by others, as well as a fair judgement of the reader. Although I am familiar with the main topic of the study, I am not an expert on the wet lab analyses carried out, so my main comments are about the analysis interpretation and potential limitations.

First, the dataset is good with a large amount of urine samples. Yet, the number of parent-offspring pairs seems relatively small (below I also ask to clarify these numbers in the narrative). I think it would be good to discuss the methodological limitations of having low number of pairs in trying to answer questions about heritability. Are there other references validating similar models across sample sizes? In other words, can the study be underestimating the contribution of genetic effects? This is important as several other references cited by the authors suggest the opposite.

We thank the reviewer for their positive feedback and suggestions. Regarding the dataset limitations, as noted in line 591-592 of the Discussion, despite using one of the largest (in terms of number of individuals and the years of data collection) cortisol datasets of its kind in a wild, long-lived mammal population, our pedigree is shallow compared to studies in shorter-lived species. We agree it is right to highlight this limitation and now note the pedigree sizes of other glucocorticoid heritability studies in lines 588-596. While there are studies with similar sized datasets (e.g., Bairos-

Novak et al., 2018 – full reference in main manuscript), in lines 599-602, we now highlight that our Bayesian approach, allied with permutations of the pedigree, allow us to account for the uncertainty in our heritability estimates, and can give confidence that our results are not strictly the result of the small pedigree structure. We now also make more explicit in lines 607-610 of the Discussion that we do not believe that genetics have no role in variation in cortisol levels in chimpanzees, especially considering results in other species. Instead, the important finding in our study is that non-genetic maternal effects are clearly larger in size than genetic effects, and that between-group differences are the main source of variation in urinary cortisol in this dataset.

On the other hand, the authors mention they discarded offspring whose mother died in their analysis. I understand why they did it and I think it is a good approach. Now, it would be interesting to discuss/predict whether the contribution from between-group effects and even the contribution from group-year effects found has something to do with group-specific demographic rates (i.e., sex- and age-specific mortality). For example, if a portion of the population structure in each site presents higher mortality risks, do the other members (those in the analysis) respond to this in term of their cortisol levels? Is the social structure after an individual dies significantly disrupted and could this affect cortisol variation at the community level? The authors mention potential ecological stressors but none of them address demographic rates and potential social disruption directly. On this same topic, can other dimensions of sociality (social integration, connectivity) be a major contributor of the observed cortisol variation, rather than just rank? I think mentioning this is worth and can lead to future questions.

This is an excellent point and suggestion. Indeed, we see from some of the fixed effect results, namely community size (Table S2 in supplementary materials), that demography is related to variation in urinary cortisol values in this species. We now specifically highlight the potential importance of demography, the loss of key individuals, and overall social cohesion in lines 490-506 of the Discussion.

Below minor comments for each section of the manuscript.

Introduction:

I don't think the first paragraph needs to be separate from the second. To me, it would read better if they were together.

The two paragraphs have been combined.

Lines 62-63. This is very interesting and important for your claim. I suggest you add one or two other references from other primate populations to make the statement stronger. I know there are several examples from the Cayo Santiago macaques. There are probably other references from other populations as well.

We have added additional primate studies to the references here (lines 60-64)

Lines 89-93. This sentence is a bit complicated. I suggest splitting it into two.

This has been done (line 89-94)

Materials & Methods.

Lines 687-690. Is this health data collected as part of the systematic data? If yes, clarify this in lines 618-620. If not, clarify here how this was collected.

We have clarified this (line 730-732)

Line 771. If ID is the identity of the individual, then is the subscript “individual” the same as “ID”? I find it a bit confusing to use two different symbols for the same variable. Clarify.

Thank you for noticing this. On lines 814-817 we state the variance components and are now consistent in the terminology used throughout this section, i.e., we use $V_{\text{individual}}$ for the variance in urinary cortisol concentrations explained by individual identity across years, and $V_{\text{ID-year}}$ for the variance explained by the ID-year variable, used to measure within-year variance in urinary cortisol concentrations as compared to the between .

Results.

Lines 177-181. You mention 310 offspring with mothers and 185 with father and mother. Then you reduce the total sample size to 170. In Table 1, you mention that 31 offspring had their mother sampled, 9 their father, and 33 both parents. Can you clarify here (after mentioning the 310 offspring and 185 offspring) how many parent-offspring were finally included in the analysis?

We have added a line to make clear that there were 64 mother-offspring and 42 father-offspring pairs in the whole study (lines 183-184).

Lines 243-244. Can you clarify here that you did not include random slopes for individual identity?

This has been done (lines 244-250).

Discussion.

See my main comments above.

Reviewer #2 (Remarks to the Author):

In this paper, the author(s) use an impressive sample size ($n=170$) for a wild-long lived primate to demonstrate that circadian stress phenotype, measured from urinary cortisol assays, shows some consistency within individuals but differs between groups. Moreover, the authors provide evidence that maternal effects but not genetics are associated with differences in within group stress phenotype. They conclude that community and maternal effects drive plasticity in stress phenotype. The authors rigorously assess the robustness of their main findings using several strategies. First, based on longstanding work in this system, they have substantial prior knowledge to inform their models, and thus, control for potential confounding. Next, they use sensitivity analyses, including an analysis of single social group and in models that include dominance rank to assess the stability in the direction, magnitude, and precision of their estimates of interest. Finally, they use permutations of the

animal IDs that generated random genetic and maternal relationships to assess possible type I error. I found the analysis to be thoughtful and rigorous. I also wanted to note that I think the authors' description of the distinction between traits having a genetic basis and being heritable (lines 134-148) is important, clear, and likely of great use to potential readers before they view the results. Finally, the paper is well written, to the point that many of the questions I formulated when reading through the results were already addressed in the discussion. This was a joy to read. In addition to these broad sentiments, below I pose a few questions and provide some (minor) suggestions that I hope improve the manuscript.

We thank the reviewer for this very positive feedback and their helpful comments which we address in detail below.

Questions:

1. Given the large proportion of variance explained by community, or socioecological macroenvironmental conditions, could the authors provide some information and discussion about 1) dynamics of social hierarchies in this species and 2) possible effects of discordant timing of sample collection. More specifically, it is noted in the introduction that individual social phenotypes are stable (lines 119-120), but are dominance hierarchies relatively stable in these groups when the data were collected? Controlling for community size and the sensitivity analysis, in which rank is included in the model is useful and may be the best solution given the data, but whether being high or low rank is stressful per se likely depends on the context in which that rank is achieved. Second, could between group differences reflect the timing of when data were collected since data collections were, to some extent, non-overlapping? Even among the Tai groups the study periods are variable. Is it possible that there are cohort effects that drive the large between group differences in variance and is there any concern that this may cause an underestimation of the genetic effect?

Thank you to the reviewer for raising these questions. Regarding the stability of dominance hierarchies, we do know that in Tai, instability in male hierarchy causes increases in urinary cortisol levels in male chimpanzees, but across all males, rather than those either lower or higher ranking (Preis et al., 2019).

In Tai and during the study period, the mean (\pm SE) tenure alpha tenure length of a male is 3.14 ± 0.48 years; a change in alpha male occurs in 0.22% of all study years; and only in one year and one community has there been more than one alpha male in a study year (South group in 2015 in which there were three different alpha males at different points). In Budongo, the tenure lengths seem longer: the one tenure we within our study period that we could determine (i.e. had a defined beginning and end) was 7 years.

Given that these unstable periods are relatively rare and typically short-term (lasting no more than 5 months in Tai for example), to our mind, they are unlikely to explain the longer-term group differences seen in our study. In line with your comments and those of reviewer 1, in the Discussion we now acknowledge that there are other social dynamics, such as levels of cohesion generally, that remain unexplored as drivers of stress that could help explain the results observed in our study (lines 490-506).

Regarding the degree of overlapping data, this is an interesting point, but we would contend that our model structure, which includes temporal components at the point of collection (ID-year, group-year) and at the point of hormonal analysis (the project/database code and LCMS method variables), should be accounting for this. Instead, we see a smaller contribution of the yearly variation in group differences,

compared to the long-term group effect. Furthermore, in 95% of the study months included in the study, more than one community was sampled.

To further test potential temporal batch effects, we ran additional models using a year-month random effect. We found that this temporal batch effect accounted for only a small part of the variance (~0.3% of the intercept, with confidence intervals 0.01-1.4%), and all the reported results held unaffected. For example, the strong effect of groups in explaining the average variance (intercept) still accounts for 96.7% of the variance.

We also tested this model while restricting our analyses to only data points for which data were available for more than one population. Results were almost identical: ~0.3% of the linear slope was accounted for by the month-year variance, with confidence intervals 0.01-1.5%; 97.4% of the intercept variance of the slope was explained by group effects.

We found minor variations in these variations of the model. For instance, we saw a point estimate for the variance explained by between-group effects in regard to the linear slope (~30%). However, all estimates remain within the confidence intervals and patterns do not change qualitatively, hence we decided to retain the models so far presented in the paper, and we did not find evidence that batch effects inflate our estimates of the prominent role of group effects. The new model results are mentioned in lines 794-796 of the main manuscript and presented in Table S14 of the supplementary materials.

2. I found the results in which maternal ID accounted for a non-zero variance in stress phenotype fascinating. Considering that maternal loss does not appear to have an enduring effect on adult stress phenotype and that after controlling for rank maternal identity still accounts for some of the variance in stress phenotype, do the authors have other ideas about what may underly this pattern...shared social networks between mothers and offspring or behavioral differences in maternal care? They briefly address this in the discussion on lines 557-562, though I thought it would be nice to expand on these ideas as space permits e.g., do offspring of more strongly connected mothers have lower or less reactive stress phenotypes in this species? Slightly more explanation here could help tie the results back to concepts about maternal effects and stress phenotype development in captive rodents and primates that appear in the introduction.

This is a great point and helps link our Introduction and Discussion more tightly. We have added some references into the Discussion about chimpanzee and rodent maternal styles, as well as the evidence in rodent studies about how maternal styles are related to stress physiology (lines 573-583).

3. The lack of variance in stress phenotype explained by genetics is also interesting, especially considering prior work in human twin studies. In reconciling these discrepant findings, the authors note that human studies often lack the same detail in measuring environmental factors as studies of wild animals. Do they have any thoughts about how the assignment of paternity to 43% of the study sample may, if at all, relate to differences in variance explained by genetics?

Thank you for highlighting this issue, it is definitely something we should have initially acknowledged, and has also been highlighted by reviewer 1. As we wrote in our response above, we are confident that our Bayesian approach (which allows us to

effectively measure uncertainty in our heritability estimates) coupled with the permutations analyses (which suggest that the pedigree size and structure alone are not driving our results) allows us to effectively analyse what is a relatively shallow pedigree. Certainly, we are sure of the qualitative element of our results, i.e., socioecological and maternal effects having substantially more influence on cortisol levels than genetics. We emphasize however, that we do not believe that genetics does not play any role in chimpanzee cortisol levels. We concede, and now state in the Discussion (lines 608-610), that genetics probably do play a role in chimpanzee cortisol levels, but that the magnitude of this effect is likely overestimated when other socioecological factors are not accounted for. The unique nature of our dataset allows us to test socioecological and maternal effects in a way that arguably has not been done in studies where genetics have made strong contributions to cortisol regulation.

Minor Comments

1. Line 86: Is describing parental care as 'short in duration' appropriate considering the species-specific life histories and given that some small animals have relatively quick development and short lifespans? In either case, I don't think post-hatching/post-parturition parental care is absent in any of the species studied in the references cited.

This is a great point; we have removed this phrasing (lines 86-87).

2. Lines 98-101: For a more nuanced presentation of the maternal effects on GR DNA methylation and downstream stress physiology, the authors may consider citations that failed to reproduce the work of Meaney et al., i.e., where low licking and grooming is associated with higher GR DNA methylation. For example, Fleming et al., (<https://doi.org/10.1037/bne0000014>) find high licking and grooming is associated with higher GR DNA methylation.

Thank for highlighting this study – we were not aware of these results and now include it in our references.

3. Some of the italicized text in table 4 does not match the table description. For example, the CI for ID-year estimate for quadratic slope is not italicized but the ICI is 0.01. Also, do the authors have a specific threshold for 'large range' CI used to distinguish bold vs italicized text?

Thank you for spotting these errors. We now also provide a definition for "large range" for the CIs, i.e., CI ranges in excess of 80% (Table 4 legend).

4. For Figure 1, I note a few minor points that may help improve interpretability. 1) For the within-group panel b, having the explanatory variables listed in the same order as in table 4, would be useful. 2) There appears to be differences in the ICI error bars even when those are supposed to equal 0. Maybe this is a visual artefact of image quality or possibly a null reference line may help here. 3) Table 4 and Figure1 appear redundant. Personally, I found table 4 to be more informative.

5. For figure 2, the dashed line for the median observed variance is hard to read, especially where it intersects the distribution of estimates from the permutations.

Thank you for the suggested changes to the figures, which have been incorporated in the revised manuscript. While we see your point regarding Figure 1 and Table 4, as the journal allows space for both, we are keen to keep both available to provide full details and a visualisation of these key results.

6. Lines 490-494, this is an interesting negative association between group size and cortisol. While not critical, I'd be interested to know if dominance hierarchies are more stable in larger groups?

As noted in our response to your broader comment above – there have been periods of rank instability in Tai, with this instability related to cortisol levels in other studies. We have incorporated some sentences on this in the Discussion (lines 491-506).

Reviewer #3 (Remarks to the Author):

This paper aims to determine the contribution of genetics, parental effects, and the socioecological macroenvironment to variability in cortisol phenotypes in populations of chimpanzees over 18 years. The major claims of the paper are that the largest proportion of variability is explained by the macroenvironment and secondary to that, the microenvironment. This is novel and exciting data for the field. As the authors rightly pointed out, these studies have not been done in a long lived species with protracted parental care. These studies have not been carried out in humans either without some extreme conditions that invariably will have major influence on the HPA axis. Given this, the contribution of genetics may have been overestimated in previous studies. As someone who has studied the HPA axis and stress reactivity throughout my career, this paper will affect how I think about the phenotypic variability associated with cortisol as we look at the relationships between parents and their children. The manuscript is logical and well written. I have only minor suggestions.

We are very grateful for the positive feedback from the reviewer, and we are very happy to hear that our paper will impact perspectives on stress reactivity.

Discussion

Lines 585-588 - I am not convinced that the urine correction (specific gravity vs. creatinine) between the current study and Sonnweber et al., 2018 would lead to different results as the majority of studies have shown that these two measures are highly correlated.

Thank you for this comment. We agree that the urine correction alone is unlikely to explain the discrepancies between the current study and Sonnweber et al., 2018. Indeed, the cortisol concentrations corrected for SG are correlated with those corrected with creatinine in our dataset, although quite weakly: $\rho = 0.31$ across whole dataset, $\rho = 0.34$ among adult males only (the demographic of focus in Sonnweber et al., 2018). When using an SG correction, samples are corrected using the population mean for SG (line 687). This can reduce variation across samples compared to creatinine correction which is not based on a population mean. This reduced between sample variation in turn might limit differences in calculated slopes. Therefore, we think it is the combination of the factors highlighted, rather than any single factor alone, including the correction method, that has led to the discrepancy. We've made this clearer in the Discussion (line 618-626).

I am fascinated that this paper shows that socioecological effects have a greater impact on

cortisol phenotypic variation compared to parental factors. While some examples of these factors are presented in the discussion, I am wondering if the authors can speculate how these macro environmental effects could potentially translate to humans since the ecological conditions are not present in the same way as in the chimpanzees.

This is an interesting point, we have expanded slightly on the relevance to human populations in lines 538-544 and 640-646 of the Discussion.

REVIEWERS' COMMENTS:

Reviewer #1 (Remarks to the Author):

I have now read the revised version of the manuscript by Tkaczynski and colleagues. I am pleased with their revision as it was thorough. I agree with the other referees' comments about the significant contribution that this study will have on the field. I have no further comments.

Reviewer #2 (Remarks to the Author):

Dear Editor,

I have read through the authors' point-by-point response and the revisions that they made to the manuscript titled, 'Shared community effects and the non-genetic maternal environment shape cortisol levels in wild chimpanzees.' I found that the authors were thorough and thoughtful in their responses to my questions and minor suggestions. In particular, I found the text that they added in the discussion regarding social dynamics to be valuable. I also appreciate that the authors further tested the rigor of their results by running additional models. Finally, I found that the authors' revisions present an even-handed discussion of the genetic vs environmental contributions to the stress phenotype. In summary, I have no further suggestions and hope to see this paper in its published form.

Response to reviewers

24th April 2023

Dear Reviewers,

Thank you for the positive feedback and approval of the final version of the manuscript. Your comments have been extremely helpful throughout this review process.

Sincerely,

Drs Patrick Tkaczynski & Fabrizio Mafessoni